# Alpha, Beta and Gamma Taxonomy of Biocontrol Agent *Diaeretiella rapae* (Hymenoptera, Braconidae)

**DOI:** 10.3390/insects16070736

**Published:** 2025-07-18

**Authors:** Nemanja Popović, Korana Kocić, Željko Tomanović, Andjeljko Petrović

**Affiliations:** 1Institute of Zoology, Faculty of Biology, University of Belgrade, Studentski Trg 16, 11000 Belgrade, Serbia; korana.kocic@bio.bg.ac.rs (K.K.); ztoman@bio.bg.ac.rs (Ž.T.); andjeljko@bio.bg.ac.rs (A.P.); 2Serbian Academy of Sciences and Arts, Knez Mihailova 35, 11000 Belgrade, Serbia

**Keywords:** *Aphidius*, taxonomy, morphology, phylogenetics, COI

## Abstract

*Diaeretiella rapae* is a widespread aphid parasitoid that targets various aphids, especially those found on cruciferous plants and cereals. Since 1855, the species changed its taxonomic position many times. In this study, specimens of *Diaeretiella* were analyzed using both molecular and morphological methods. The analysis of mitochondrial COI region revealed 23 different haplotypes within the genus. Phylogenetic analysis placed *D. rapae* within the *Aphidius* clade, with genetic distances that are within the range of intraspecific genetic distances of *Aphidius*. Morphological comparisons showed that *D. rapae* shares key morphological characters with species in the genus *Aphidius*. Based on the results, we synonymize *D. rapae* as *A. rapae* and designate genus *Diaeretiella* as junior synonym of *Aphidius*.

## 1. Introduction

*Diaeretiella rapae* (McIntosh, 1855) is a koinobiont endoparasitoid that parasitizes around 100 species of aphids found on more than 180 plants [1,2]. The most common hosts are *Brevicoryne brassicae* (Linnaeus, 1758), *Myzus persicae* (Sulzer, 1776), *Lipaphis erysimi* (Kaltenbach, 1843), and *Diuraphis noxia* (Mordvilko, 1913), preferentially on cruciferous plants and cereals [1,2,3]. Although it is assumed that the place of origin of *D. rapae* is the Mediterranean part of Europe, it is almost cosmopolitan in distribution, inhabiting all continents except Antarctica [2]. Due to its role as a biological control agent, this species is considered to be of high economic importance and has been well studied compared to other members of the Aphidiinae subfamily; there are numerous studies which examined its biocontrol potential [4,5,6,7,8], phylogeography [9] and its biology [10,11,12,13,14,15]. Despite broad knowledge that was gathered over the last few decades, there are still some unsolved taxonomic problems. The authorship of this species has been problematic, as it was first described in Charles McIntosh’s “The Book of the Garden” [16] in 1855 and five years later in John Curtis’ “Farm insects” [17]. Historically, most authors considered Curtis to be the author of *D. rapae* in his book or in McIntosh’s book, as Curtis was a renowned entomologist in his time, while only some people ascribed authorship to McIntosh. The problem was solved by Mackauer [18], in which Charles McIntosh was given as the author of the species according to the International Code of Zoological Nomenclature.

Since its description, *D. rapae* has changed its taxonomic position several times. Gahan [19] placed the species in the genus *Diaeretus* and separated it from the other *Trioxys* Haliday, 1833 by the absence of prongs and from *Lipolexis* Förster, 1862 by the presence of a second discoidal cell. He also noted that Marshall [20] placed all species in the genus *Aphidius,* not accepting Foerster’s genus table. Starý [21] placed *D. rapae* in a separate monotypic genus *Diaeretiella*, differentiating it from the other Aphidiinae by the reduction of the wing venation and the shape of ovipositor sheath and separating it from *Diaeretus* by the narrow central areola of propodeum. Mackauer [22] initially did not accept Stary’s classification of the species and kept it in the genus *Aphidius*, but later in the same year, *D. rapae* was accepted as a valid name [23]. Hafez [10] agreed with Mackauer’s first statement that the species belongs to the genus *Aphidius* because the reduction of the wing venation cannot be a taxonomic character in the Aphidiinae (it is present in several independent lineages), and the female genitalia of *D. rapae* are typical of the genus *Aphidius*. Up to present times, its status was unchanged, and it has been re-described twice in the previous decade [24,25]. The diagram with synonyms and their publication time is given in Figure 1.

**Figure 1 insects-16-00736-f001:**
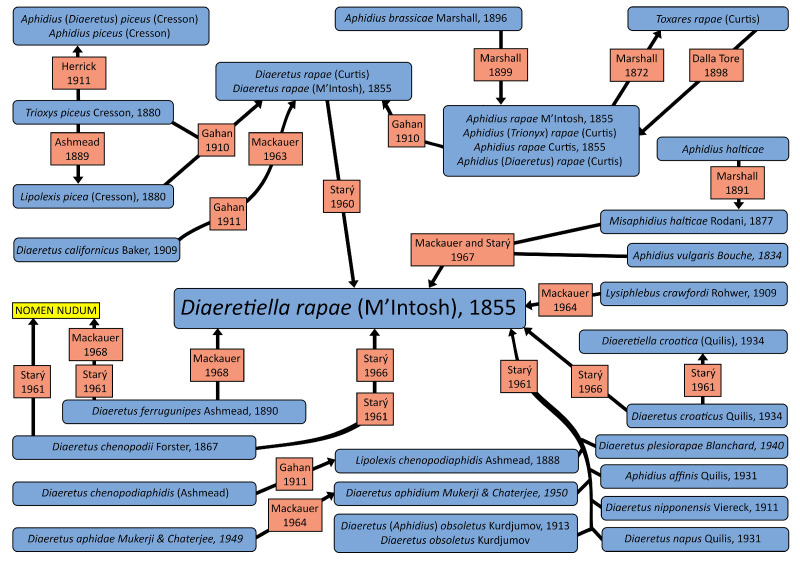
Diagram of *D. rapae* synonyms (blue—synonyms, red—publications where taxon status changed, yellow—nomen nudum) [18,19,20,21,26,27,28,29,30,31,32,33,34,35,36].

The main goal of this research is to resolve the taxonomic position of *D. rapae* using morphological and molecular analysis.

## 2. Materials and Methods

### 2.1. Sample Collection and Morphological Analysis

Specimens were collected between 1989 and 2023 from sites across Europe and the Middle East (Serbia, Montenegro, Slovenia, Greece, Italy, Belgium, Russia and Turkey). Parasitoids were collected by sampling plant parts infested with aphids. Plants with aphid colonies were placed in plastic containers covered with a nylon mesh to allow ventilation. The samples were transported to the laboratory and kept under controlled conditions (22 °C, 65% relative humidity, 16 h light/8 h dark). After emergence, the parasitoids were preserved in 96% ethanol or dry mounted. Aphid and plant samples were identified to genus or species level to establish tri-trophic interactions. In total 1741♀ and 1145♂ belonging to *D. rapae* were examined under ZEISS SteREODiscovery.V12 (ZEISS Microscopy, Jena, Germany). Specimens were photographed using the Leica DM LS phase contrast microscope (Leica Microsystems GmbH, Wetzlar, Germany), while relevant measurements were taken using the ImageJ 1.50i software [37].

List of all *D. rapae* specimens is given in Appendix A. In addition, we examined specimens belonging to 43 *Aphidius* species listed in Table 1, as those exhibit the entire morphological variability in the genus *Aphidius*. Several *Diaeretiella* and *Aphidius* specimens were sputter coated with gold and examined using a Jeol JSM-6460LV scanning electron microscope (Jeol Ltd., Tokyo, Japan). For inter-generic comparison, we analyzed the following characters commonly used for *Aphidius* species [38]: antennae—number of antennomeres, shape of antennae, length of flagellomere 1 (ratio between length and width of flagellomere 1 at median level), color of flagellomere 1, number of longitudinal placodes on flagellomeres 1 and 2; number of labial and maxillary palpomeres; tentorial index (ratio between tentoriocular line/intertentorial line); forewing venation (length/width of pterostigma, ratio between length of vein R1 (= metacarpus) and length of pterostigma); petiole-dorsal and anterolateral sculpturation, length of petiole (ratio between length and width of petiole at the spiracle level); propodeal areola (closed or open); shape of ovipositor sheath (Figure 2). The terminology of morphological characters follows Sharkey and Wharton [39]. The parasitoids examined in this study are deposited in the collection of the Institute of Zoology, Faculty of Biology, University of Belgrade (FBUB).

**Table 1 insects-16-00736-t001:** List of *Aphidius* species used in this study.

*Aphidius absinthii* Marshall, 1896	*Aphidius matricariae* Haliday, 1834
*Aphidius aquilus* Mackauer, 1961	*Aphidius microlophii* Pennachio & Tremblay, 1987
*Aphidius areolatus* Ashmead, 1906	*Aphidius phalangomyzi* Starý, 1963
*Aphidius arvensis* (Starý, 1960)	*Aphidius platensis* Brethes, 1913
*Aphidius asiaticus* Kim & Tomanović, 2021	*Aphidius plocomaphidis* (Starý, 1973)
*Aphidius avenae* Haliday, 1834	*Aphidius rhopalosiphi* de Stefani-Perez, 1902
*Aphidius avenaphis* (Fitchk, 1861)	*Aphidius ribis* Haliday, 1834
*Aphidius balcanicus* Tomanović & Petrović, 2011	*Aphidius rosae* Haliday, 1834
*Aphidius banksae* Kittel, 2016	*Aphidius salicis* Haliday, 1834
*Aphidius cingulatus* Ruthe, 1859	*Aphidius schimitscheki* (Starý, 1960)
*Aphidius colemani* Viereck, 1912	*Aphidius setiger* (Mackauer, 1961)
*Aphidius eadyi* Starý, Gonzáles & Hall, 1980	*Aphidius smithi* Sharma & Subba Rao, 1959
*Aphidius ericaphidis* Pike & Starý, 2011	*Aphidius sonchi* Marshall, 1896
*Aphidius ervi* Haliday, 1834	*Aphidius staticobii* Tomanović & Petrović, 2012
*Aphidius funebris* Mackauer, 1961	*Aphidius sussi* Pennachio& Tremblay, 1989
*Aphidius gerani* Tomanović & Kavallieratos, 2009	*Aphidius tanacetarius* Mackauer, 1962
*Aphidius gifuensis* Ashmead, 1906	*Aphidius tarsalis* van Achterberg, 2006
*Aphidius hieraciorum* Starý, 1962	*Aphidius transcaspicus* Telenga, 1958
*Aphidius hortensis* Marshall, 1896	*Aphidius urticae* Haliday, 1834
*Aphidius leclanti* Tomanović & Chaubet, 2013	*Aphidius uzbekistanicus* Luzhetzki, 1960
*Aphidius longipetiolus* Takada, 1968	*Aphidius viaticus* (Sedlag, 1968)
*Aphidius longistigmus* Kim & Tomanović, 2021	

**Figure 2 insects-16-00736-f002:**
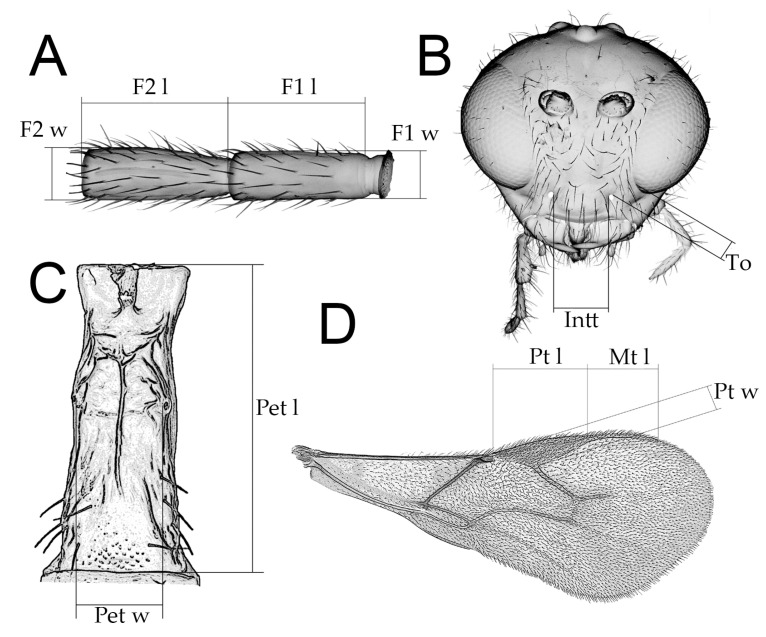
Schematic drawing of character measurements: (**A**) First and second flagellar segments, F1 l—length of F1; F1 w—width of F1; F2 l—lenght of F2; F2 w—width of F2; (**B**) Head, frontal view, Intt—lenght of intertentorial line; To—length of tentoriocular line; (**C**) Petiole, dorsal view, Pet l—length of petiole; Pet w—width of petiole; (**D**) Fore wing, Pt l—length of pterostigma; Pt w—width of pterostigma; Mt l—length of vein R1 (= metacarpus).

### 2.2. Molecular Analysis

For 25 specimens, total genomic DNA was extracted non-destructively with Qiagen Dneasy^®^ Blood & Tissue Kit (Qiagen Inc., Valencia, CA, USA), following the manufacturers’ protocol. The universal primers LCO1490 and HCO2198 were used for amplification of the cytochrome oxidase subunit I region [40]. The amplification mixture contained 32.6 μL of nuclease free water, 10 μL of amplification buffer, 1 μL of nucleotide solution, 1 μL of primers (each), 0.4 μL of taq polymerase and finally 4 μL of extracted DNA, adding to the final volume of 50 μL. The PCR amplification profile was: 5 min of initial denaturation, 37 cycles of 60 s (94 °C), 60 s annealing (56 °C) and 90 s extension (68 °C) and 7 min of final extension (72 °C). Amplification products were purified and sequenced by Macrogen Inc. (Seoul, Republic of Korea).

Electropherograms were visualized in Finch TV Geospiza Inc. (Seattle, WA, USA) and manually edited and aligned using BioEdit software 7.2.5 [41]. The analysis of evolutionary divergence for *Diaeretiella* sequences was conducted in MEGA 11 software [42] using the Kimura 2-parameter distance model. The evolutionary history of *Diaeretiella rapae* was inferred by using the Maximum Likelihood (ML) method and Tamura–Nei model [43], which was indicated by MEGA 11 as the best fitting model. The final dataset for the reconstruction of phylogenetic relationships between *Diaeretiella* samples consisted of 315 COI sequences: 25 newly acquired sequences, additional 290 sequences acquired from GenBank, and one outgroup sequence—*Toxares deltiger* (Haliday, 1833) (Appendix A). *Diaeretiella rapae* haplotype diversity was estimated by the software DNAsp v6.10.04 [44], and a haplotype network was constructed by Network (version 10.2.0.0).

To establish the phylogenetic position of *Diaeretiella* within subfamily Aphidiinae, additional 68 sequences were acquired from GenBank, representing 13 Aphidiinae genera. The final dataset for this phylogenetic reconstruction consisted of 73 COI sequences (5 newly acquired sequences).

In order to determine phylogenetic relationships between *Diaeretiella* and *Aphidius* species, three outgroup species were used, *Venturia canescens* (Gravehorst, 1829), *Ephedrus niger* Gautier, Bonnamour & Gaumont, 1929 and *Toxares deltiger*. The final dataset for this phylogenetic reconstruction consisted of 138 COI sequences (3 newly acquired sequences, additional 135 sequences acquired from GenBank, and 3 outgroup sequences) (Appendix A).

In both phylogenetic reconstructions, Bayesian evolutionary analysis was performed with BEAST 2.5 [45] software employing the initial data set constructed in BEAUti v1.10.4 [45], with designated strict clock type and Yule process of speciation. The analysis ran for 10 million generations, and the sampling was conducted every 1000 generations, while the first million trees were discarded as a burn in. The effective sample size (ESS) of the parameters of the Markov chain Monte Carlo was estimated by Tracer v1.7.1 [46]. The saturation level for the third codon position was inspected in DAMBE 7.2.133 software [47] using the Xia model test [48]. The phylogenetic tree was visualized using FigTree 1.4.3 software [49]. 

## 3. Results

### 3.1. Morphology

All analyzed morphological characters of *D. rapae* fall within the determined variability of the same characters of the genus *Aphidius*. Number of antennomeres corresponds to *Aphidius* species that possess short antennae with only 13–14 segments (*A. setiger*, *A. aquilus*, *A. salicis*). Species of the genus *Aphidius* possess variable number of maxillar and labial palpomeres (Table 2, MP, LP values), whereas *D. rapae* possesses maxillar and labial palps constituted out of 3–4 and 2 palpomeres, respectively, which is a value that is characteristic for most *Aphidius* species. Propodeum possesses a visible central narrow pentagonal areola that is also present in most *Aphidius* species. Its shape is also characteristic to most of the other *Aphidius* species. The lateral side of the petiole has curved costulae, also present in most *Aphidius* species. The shape of *D*. *rapae* ovipositor sheaths is typical for *Aphidius*. Table 2 gives the values of characters in *D. rapae* as well as the minimum and maximum range for the genus *Aphidius*. The state of some characters in both *D. rapae* and *Aphidius* spp. is shown in Figure 3, Figure 4, Figure 5 and Figure 6. For every character we chose five *Aphidius* species which reflect total variability of character states.

**Table 2 insects-16-00736-t002:** Value of morphological characters important for taxonomy of genus *Aphidius*.

	*D. rapae*	*Aphidius* (Minimal Value)		*Aphidius* (Maximal Value)	
**Antennae**	(12) 13–14	12–13	(*A*. *salicis*)	(19) 20–21	(*A. eadyi*)
**F1 l/w**	2.5–3.5	2.11–2.52	(*A*. *longistigmus*)	4–5	(*A. gifuensis*)
**F2 l/w**	2–3	1.63–1.76	(*A*. *areolatus*)	3–4,3	(*A. banksae*)
**MPS F1**	0–1	0	(*A*. *ribis*)	3–6	(*A. rosae*)
**MPS F2**	2–4	0	(*A*. *leclanti*)	4–7	(*A. cingulatus*)
**F1/F2**	1	0.85–0.93	(*A*. *banksae*)	1.06	(*A. leclanti*)
**MP**	3–4	2	(*A*. *arvensis*)	4	(*A. absinthii*)
**LP**	2	1	(*A*. *arvensis*)	3	(*A. uzbekistanicus*)
**Ti**	0.29–0.36	0.3–0.4	(*A*. *schimitscheki*)	0.74–0.8	(*A. cingulatus*)
**Pt l/w**	2.9–4	2.8–3	(*A. plocomaphidis*)	4.96–5.46	(*A. longistigmus*)
**Ptl/Mtl**	1.39–2.63	0.93–1.05	(*A*. *urticae*)	1.5–2	(*A. avenae*)
**Pet l/w**	1.93–2.65	2.17	(*A*. *areolatus*)	3.88–4	(*A. geranii*)

Antennae—number of antennomeres; F1 l/w—ratio between length and width of flagellomere 1 at median level; F2 l/w—ratio between length and width of flagellomere 2 at median level; MPS F1—number of longitudinal placodes on flagellomere 1; MPS F2—number of longitudinal placodes on flagellomere 2; F1/F2—ratio between lengths of flagellomere 1 and 2; MP—number of maxillary palpomeres; LP—number of labial palpomeres; Ti—ratio between tentoriocular line/intertentorial line; Pt l/w—ratio between length and width of pterostigma; Ptl/Mtl—ratio between length of pterostigma and length of vein R1; Pet l/w—ratio between length and width of petiole at level of spiracles.

**Figure 3 insects-16-00736-f003:**
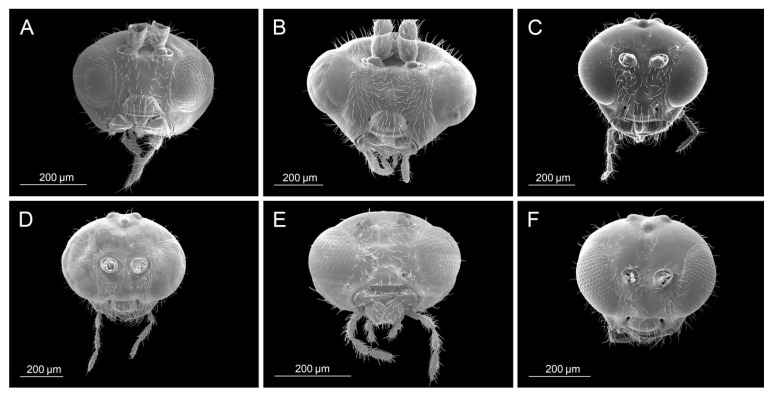
Head: (**A**) *Diaeretiella rapae*, (**B**) *Aphidius cingulatus*, (**C**) *Aphidius balcanicus*, (**D**) *Aphidius banksae*, (**E**) *Aphidius ericaphidis*, (**F**) *Aphidius platensis*.

**Figure 4 insects-16-00736-f004:**
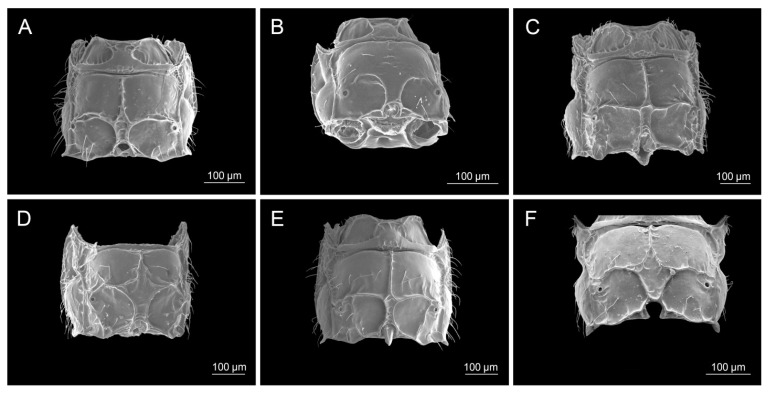
Propodeum: (**A**) *Diaeretiella rapae*, (**B**) *Aphidius staticobii*, (**C**) *Aphidius eadyi*, (**D**) *Aphidius banksae*, (**E**) *Aphidius ervi*, (**F**) *Aphidius colemani*.

**Figure 5 insects-16-00736-f005:**
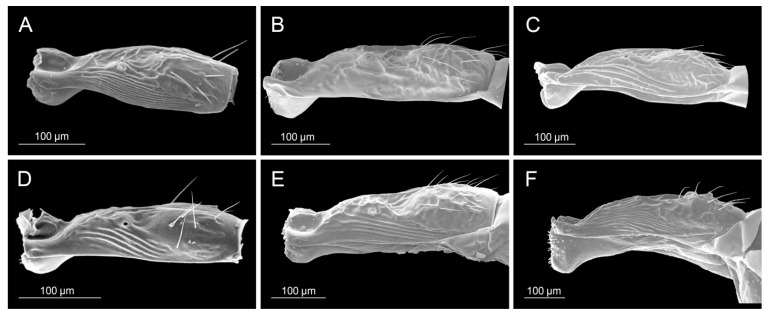
Petiole: (**A**) *Diaeretiella rapae*, (**B**) *Aphidius ervi*, (**C**) *Aphidius avenae*, (**D**) *Aphidius colemani*, (**E**) *Aphidius smithi*, (**F**) *Aphidius eadyi*.

**Figure 6 insects-16-00736-f006:**
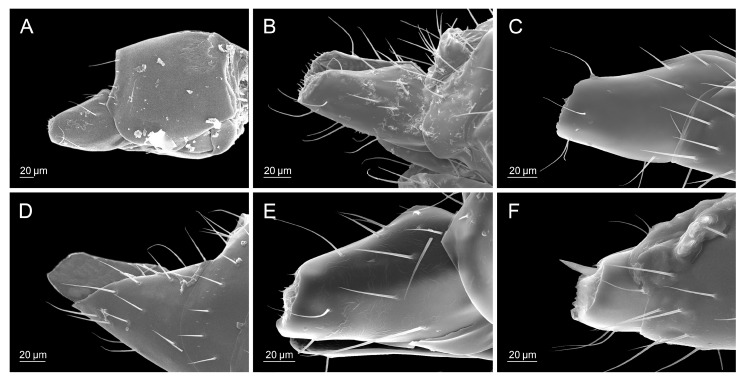
Ovipositor sheaths: (**A**) *Diaeretiella rapae*, (**B**) *Aphidius ericaphidis*, (**C**) *Aphidius geranii*, (**D**) *Aphidius staticobii*, (**E**) *Aphidius balcanicus*, (**F**) *Aphidius urticae*.

The most important character that currently discriminates genus *Diaeretiella* from *Aphidius* is reduced wing venation, with the median (M + m − cu) and r − m veins absent in *D. rapae*, which is also the case for some *Aphidius* species. For example, in *A. aquilus* it is very common to have specimens with and without M + m − cu and r − m veins present within the same population (Figure 7). Also, some populations of *A. salicis* show similar venation pattern as *A. aquilus*. Species of the subgenus *Lysaphidus* have developed only a small part of M + m − cu vein under a developed r − m vein [49].

**Figure 7 insects-16-00736-f007:**
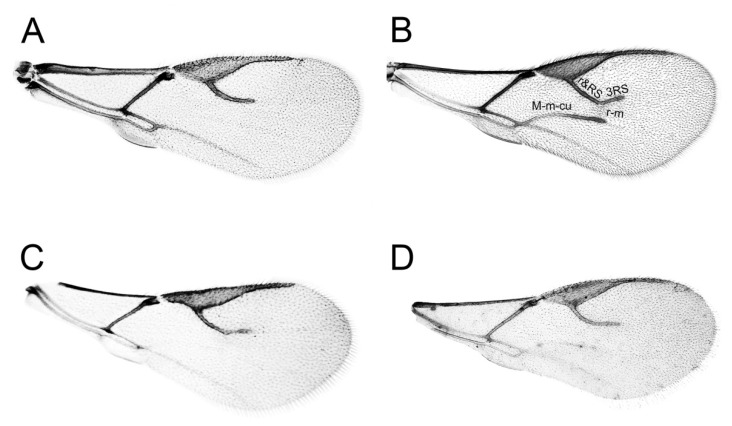
Fore wings: (**A**) *Diaeretiella rapae*, (**B**) *Aphidius ervi*, (**C**,**D**) *Aphidius aquilus*.

### 3.2. Molecular Analysis

The 315 COI sequences obtained from *D. rapae* comprised 23 haplotypes (H1–H23) with mean genetic distance of 0.8% between haplotypes. The highest recorded intraspecific genetic distance was 1.4% between haplotypes H9 and H20 (Appendix A).

The obtained ML phylogenetic tree of *D. rapae* (presented as a single sequence haplotype tree in Figure 8) is in congruence with determined low genetic diversity between haplotypes.

The reconstructed phylogenetic network shows a slight separation of haplotypes from the Indian subcontinent (Figure 9). Within this group of six haplotypes (Figure 9 right side), H7 is the most common with 62 sequences and the nodal point for the remaining haplotypes, while H8 is represented with a single sequence from India. The other 17 haplotypes (eight with a single sequence), were grouped together (Figure 9, left side) with H2 as the most abundant, containing 180 sequences from Holarctic, Neotropic and Afrotropic regions.

Analysis of phylogenetic relationships between *Diaeretiella rapae* and other Aphidiinae genera (Figure 10) revealed that *D. rapae* is nested within *Aphidius* species.

Analysis of phylogenetic relationships between *Diaeretiella rapae* and *Aphidius* species (Figure 11) revealed that *D. rapae* is nested within *Aphidius* species. According to COI sequences, *D. rapae* is a monophyletic species which together with all analysed *Aphidius* species forms a monophyletic clade. *Diaeretiella rapae* formed a separate clade with *A. viaticus*, *A. schimitscheki, A. salicis* and *A. aquilus. Aphidius viaticus* positioned as the closest relative to *D. rapae,* with a genetic distance of 2.1% between them. Those two species together form a sister clade to a clade consisting of *A. schimitscheki*, *A. aquilus* and *A. salicis,* with the mean genetic distance of 6.3% between the two clades. *Aphidius platensis* showed the highest mean genetic distance (11.3%) from *D. rapae*. Mean genetic distances within analysed *Aphidius* species varied from 0.0% to 11.7%.

There are no significant differences between *Diaeretiella rapae* and *Aphidius* species, based on both morphological and molecular analysis that would discriminate between these two genera.

### 3.3. Re-Description of Aphidius Rapae McIntosh, 1855

Type material: Original type specimen is not available and is probably lost. Mackauer [18] designated a single female of *Lysiphlebus crawfordi* as a lectotype of *D. rapae* (Phoenix, Ariz.; “*Myzus persicae*”; U.S.N.M., No. 66797). The whole specimen is slide mounted. The authors of this study examined lectotype photographs provided by Robert Kula.

Diagnosis: In most cases, *Aphidius rapae* differs from other *Aphidius* species by the absence of the M + m − cu and r − m veins. However, based only on wing venation, specimens of several species could be confused with *A. rapae*. This is because they may not have these veins or they are transparent or very pale. It is most similar to *A. schimitscheki* from which it differs by the ratio of the petiole at spiracles level (*A. rapae* 1.9–2.6; *A schimitscheki* 3.0–3.5). It can be differentiated from species of subgenus *Lysaphidus* (e.g., *A. viaticus*, *A. arvensis*, *A. erysimi*) by a developed r − m vein in *Aphidius* (*Lysaphidus*) spp. [50]. *A. aquilus* and *A. salicis* can be differentiated from *A. rapae* when specimens show slight traces of the M + m − cu vein. If this vein is not visible, these two species often exhibit a visible point of connection between the r + Rs and 3 RS veins. Additionally, the aphid hosts of *A. aquilus* and *A. salicis* typically feed on trees.

Female (Figure 12A and Figure 13)

Description: Head rounded, moderately setose. Face moderately setose. Tentorial index 0.29–0.36. Malar space 0.21–0.3 as long as longitudinal eye diameter. Mandible bidentate, covered with setae (Figure 13A). Antenna filiform, with (12) 13–14 antennomeres (Figure 13B) and with semierect setae 2/3 length of flagellomere diameter. Flagellomere 1 (F1) as long as F2 (F1/F2 1.0). F1 2.5–3.5 times as long as wide at the middle. Flagellomere 2 2.0–3.0 as long as wide at middle. F1 without or with one longitudinal placode, F2 with two to four longitudinal placodes (Figure 13C). Maxillary palp with four palpomeres, sometimes 3 (if apical segment is undivided), labial palp with two short palpomeres. Clypeus with 9–18 setae.

Mesosoma. Mesoscutum with notaulices only in short ascending part of its anterolateral area and outlined by 2 rows of long scattered setae extending almost to scutellum (Figure 13D). Scutellum with 4–10 setae laterally. Propodeum areolated, central areola pentagonal and very narrow (Figure 13F). Fore wing pterostigma 2.9–4.0 times as long as wide. Ratio between length of pterostigma and R1 vein (=metacarpus) 1.39–2.63 (Figure 13E).

Metasoma. Petiole 1.93–2.65 times as long as wide at spiracle level, with 14–18 curved costulae at anterolateral area. Petiole slightly rugose dorsally (Figure 13G).

Color. Head brown to dark brown. Mouth parts yellow. Scape and pedicel brown. Annelus and base of F1 yellowish, remainder of antenna brown. Mesoscutum dark brown to almost black. Propodeum brown. Petiole yellowish to light brown. Abdomen and ovipositor sheath dark brown. Legs yellowish to light brown. Wing hyaline.

Male (Figure 12B and Figure 14)

Morphologically similar to female.Tentorial index 0.4–0.45, malar space 0.35–0.4 as long as longitudinal eye diameter (Figure 14A). Antenna filiform, with 15–16 antennomeres (Figure 14B) and with semierect setae 1/3 length of flagellomere diameter. Flagellomere 1 (F1) as long as F2. F1 almost two times long as wide at the middle (1.93–1.95). F2 1.7–1.8 times as long as wide at middle. F1 with 4 longitudinal placodes, F2 with 5 longitudinal placodes (Figure 14C). Genitalia as in Figure 14H.

Color: Head dark brown to black. Mouth parts yellowish. Antenna brown. Mesoscutum dark brown to almost black. Propodeum and petiole brown. Abdomen dark brown. Legs light brown to brown. Wing hyaline.

Body length: 2.0 mm.

**Figure 12 insects-16-00736-f012:**
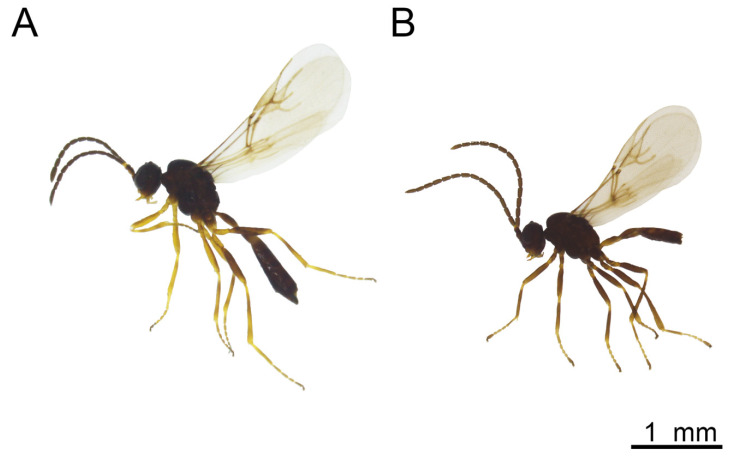
*Aphidius rapae*: (**A**) female, (**B**) male.

**Figure 13 insects-16-00736-f013:**
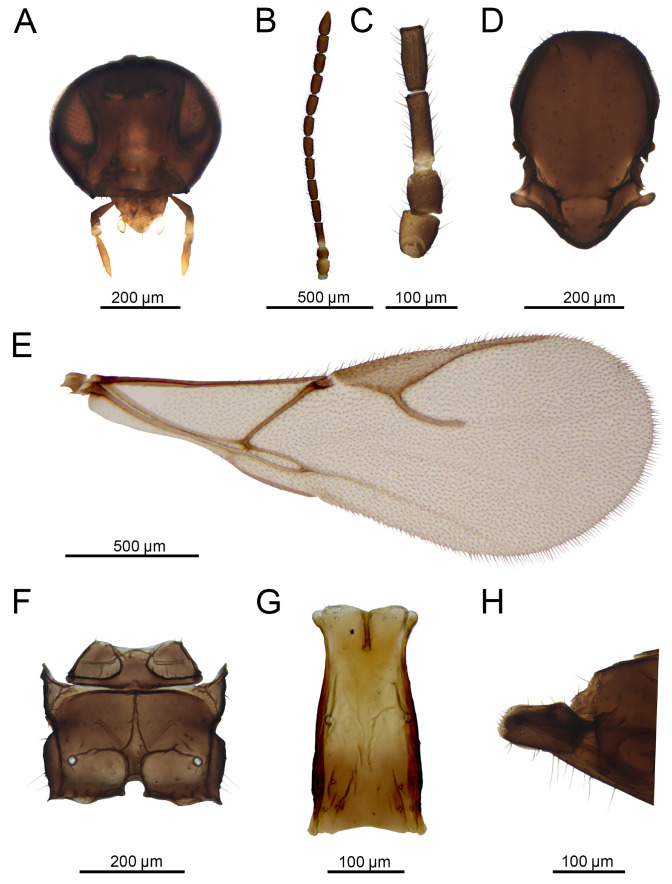
*Aphidius rapae* McIntosh, 1855 female: (**A**) head, (**B**) antenna, (**C**) scape, pedicel, annellus and first and second flagellar segments, (**D**) mesonotum (=mesoscutum + scutellum), dorsal view, (**E**) forewing, (**F**) propodeum, dorsal view, (**G**) petiole, dorsal view, (**H**) ovipositor sheath, lateral view.

**Figure 14 insects-16-00736-f014:**
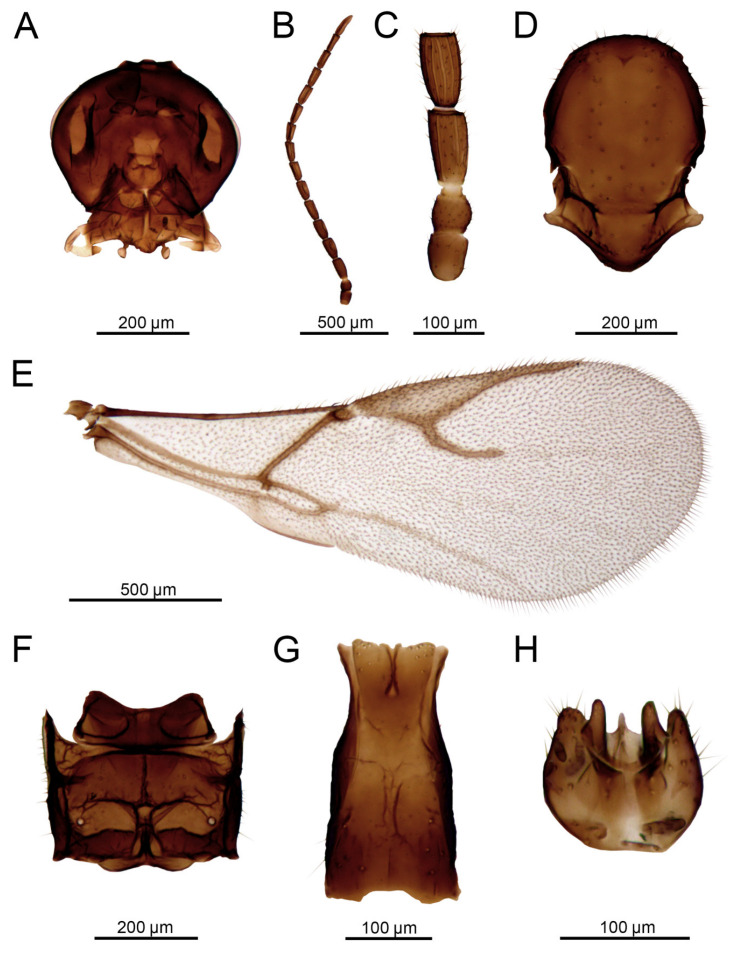
*Aphidius rapae* McIntosh, 1855 male: (**A**) head, (**B**) antenna, (**C**) scape, pedicel, annellus and first and second flagellar segments, (**D**) mesonotum (=mesoscutum + scutellum), dorsal view, (**E**) forewing, (**F**) propodeum, dorsal view, (**G**) petiole, dorsal view, (**H**) aedeagus, ventral view.

## 4. Discussion

In this study, we analyzed the taxonomic status of *A. rapae* (*D. rapae*) within the subfamily Aphidiinae and proved that *A. rapae* belongs to the genus *Aphidius* according to morphological and molecular characters. This confirms statements of Hafez [10] and Mackauer [22] that *A. rapae* belongs to this genus, as the reduced wing venation within genus *Aphidius* could not be used as a character for taxonomic determination.

As already noted, *A. rapae* is a cosmopolitan species, which is confirmed by the specimens from all over the world used in this study. The haplotype network of *A. rapae* is organized in a star shape, with the central haplotype having the most sequences, suggesting that the population of *A. rapae* has recently expanded [51]. The combination of data from the haplotype network and the phylogenetic tree of *A. rapae* confirms that most branches are monophyletic. The central haplotype (H2) is the most important node in the haplotype network, linking nine other groups/haplotypes. It has a cosmopolitan distribution (17 countries around the world) and is the most common in all samples. Although the majority of specimens (≈ 75%) were sampled from three countries (Canada, France, Serbia) because the research groups are based in these countries or extensive research has been conducted there, it does not diminish the significance of this haplotype. In the haplotype network, some haplotype patterns reflect their geographical distribution: one cluster consists exclusively of Indian specimens (H10, H11), one of specimens from Montenegro (H5, H6), one of specimens from the Balkan Peninsula (Slovenia, Montenegro and Serbia, haplotypes H3 and H4) and one haplotype from Australasia (H17). The Chinese group (H13, H14, H18) is linked to the East Asian group (H12), probably because they are closely related. These clusters are linked to the Canadian group (H15, H20), which can be explained as the cause of the introduction of *A. rapae* to North America from China in the early 1990s [52]. There is a slight separation of the haplotypes from South Asia (Pakistan and India) (H1, H7, H8, H9, H21, H22) from the others, which is also confirmed by the phylogenetic tree, suggesting a common ancestor. The results of research on the phylogeography of this species from the Old and New World show, as in our research, that *A. rapae* has recently expanded its range. This study also shows that *A. rapae* does not show genetic isolation, even when the distribution of the samples is taken into account [9]. This indicates that the genetic differences between the analysed populations in our study are not significant.

It is known that *A. rapae* was introduced to North America to control the aphid *D. noxia* because the native parasitoid population was unsuccessful in controlling this pest [52,53,54]. It was also introduced from Queensland and New South Wales to Western Australia in 1902 and from Sri Lanka in 1909 for biological control of *B. brassicae* [55]. For New Zealand, there is no record of deliberate introduction of *A. rapae*, and the first finding dates back to 1930 [56]. The separation of haplotype 17, which consists only of specimens from Australia and New Zealand and no other haplotype from these areas, gives us an indication that *A. rapae* from this region may have a different migration history. It is also shown by research based on microsatellites where the populations from Australia, Kazakhstan, France and Morocco were compared [5]. The results of this research indicated that *A. rapae* from Australia probably originates from one population as it exhibited significant differences from three other countries.

In the phylogenetic tree, *A. rapae* is positioned in the central section. It is grouped with other species, such as *A. viaticus*, *A. salicis*, *A. aquilus* and *A. schimitscheki*. This species group has a smaller number of antennomeres and various degrees of forewing vein reduction, a morphological trait that is also characteristic for *A. rapae*. The genetic distance between *A. rapae* and other *Aphidius* species ranges from 2.1 to 12.6%, while the mean distance between other *Aphidius* species is 0.0–12.9%. This also confirms that *A. rapae* is no more genetically different from other *Aphidius* species than they are from each other.

There is a general trend within the Parasitica group of Hymenoptera toward a reduction in body size and wing venation [57,58,59]. Wing venation reduction at a degree which is present in *Aphidius rapae* is also present in several other genera (*Adialytus* Förster, 1862; *Binodoxys* Mackauer, 1960; *Betuloxys* Mackauer, 1960; *Trioxys* Haliday, 1833; *Diaeretus* Förster, 1863; *Lipolexis* Förster, 1863; *Monoctonus* Haliday, 1833) and this event has occurred several times in the Aphidiinae [57]. In the genus *Aphidius*, it has probably occurred only once, which is also indicated by the results of our study. The genus *Aphidius* belongs to the more evolutionarily recent genera within the subfamily Aphidiinae, where the evolutionary trend is toward a more elongated body, a more slender petiole, reduced wing venation, and longer antennae with a greater number of antennal segments.

## Figures and Tables

**Figure 8 insects-16-00736-f008:**
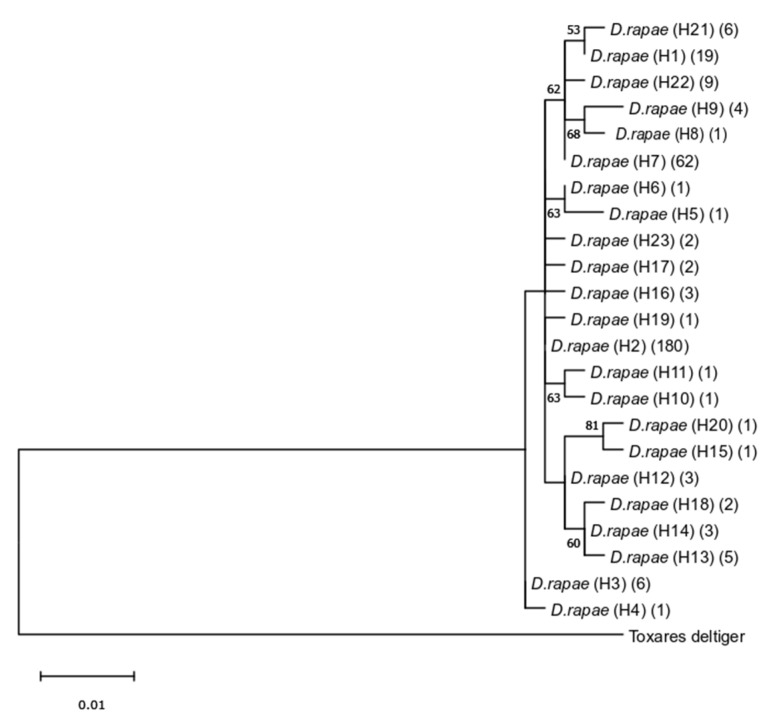
Phylogenetic tree based on partial mtCOI sequences of *Diaeretiella rapae* obtained using the Maximum Likelihood method. Bootstrap values are indicated above/below branches. Numbers in parentheses refer to the number of sequences for each haplotype.

**Figure 9 insects-16-00736-f009:**
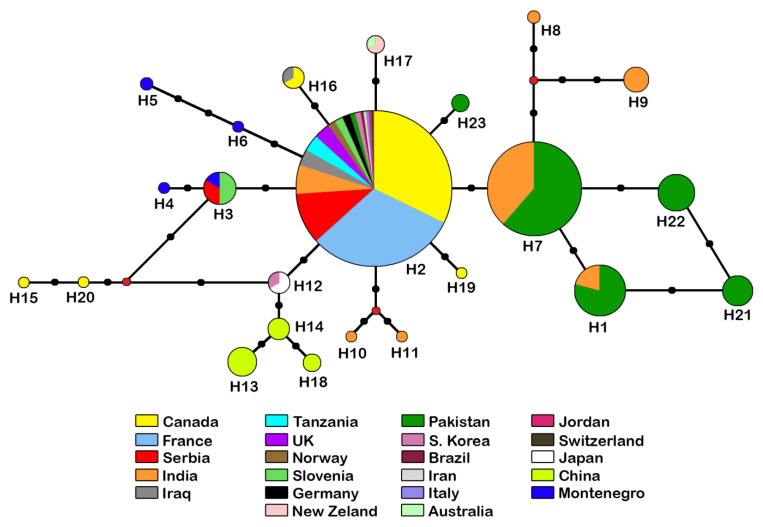
Haplotype network for COI sequences from 315 specimens belonging to the *D. rapae*. The circle size indicates the number of specimens with a haplotype (not to scale); each black dot represents a nucleotide substitution; red dots represent median vectors.

**Figure 10 insects-16-00736-f010:**
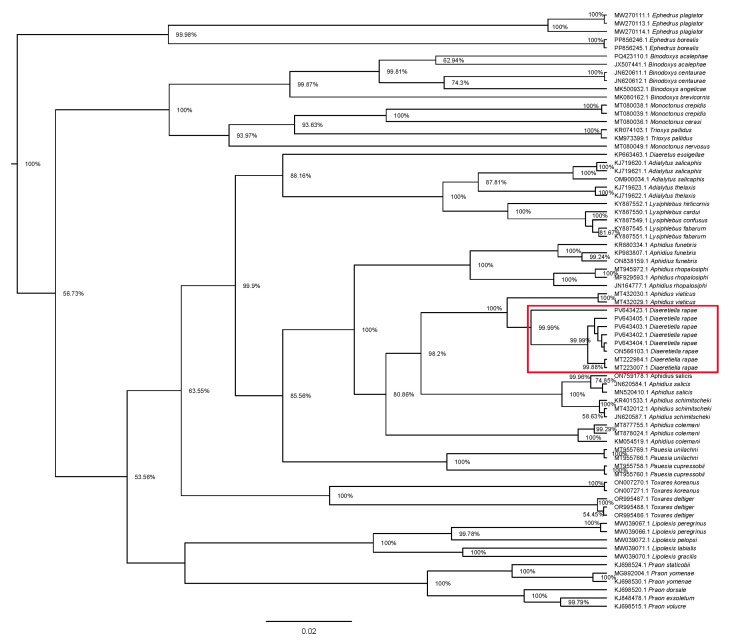
Phylogenetic tree based on 73 cytochrome oxidase *c* subunit I (COI) sequences of Aphidiinae species, representing 13 genera. Red rectangle represents *D. rapae* sequences.

**Figure 11 insects-16-00736-f011:**
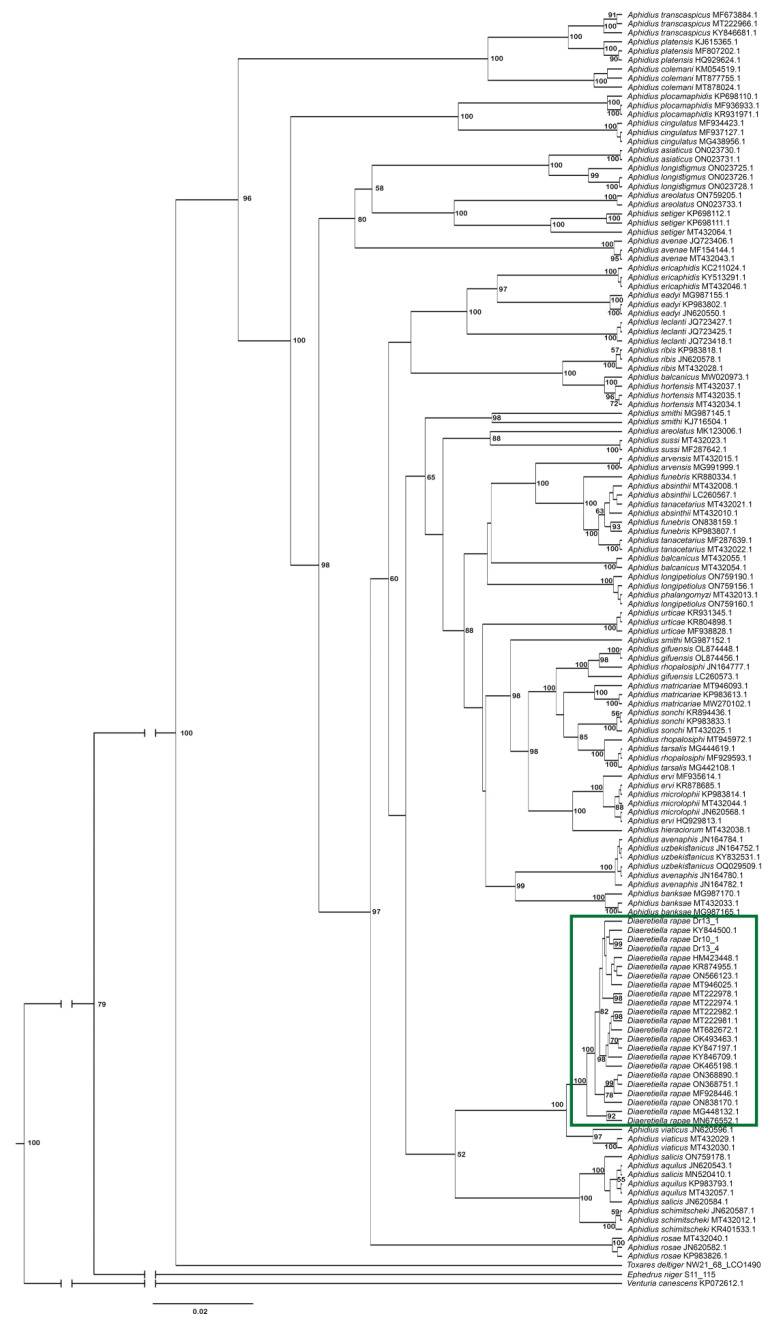
Phylogenetic tree based on 138 cytochrome oxidase *c* subunit I (COI) sequences of *Aphidius* species. The green rectangle represents *D. rapae* sequences.

## Data Availability

The new specimen sequences analyzed in this study are deposited in the GenBank (https://www.ncbi.nlm.nih.gov/genbank/) under accession numbers PV643399–PV643423.

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
