# Peer review of "Alpha, Beta and Gamma Taxonomy of Biocontrol Agent Diaeretiella rapae (Hymenoptera, Braconidae)"

_insects, 2025, doi:10.3390/insects16070736_

Round 1

Reviewer 1 Report

Comments and Suggestions for Authors

Dear Dr. Popović,

I have carefully read your manuscript entitled "Alpha, beta and gamma taxonomy of biocontrol agent Diaeretiella rapae (Hymenoptera, Braconidae)". The paper deals with the taxonomic position and intraspecific structure of this economically important species, which is considered by the authors as a member of the genus Aphidius. The manuscript is well-written and illustrated. It contains the results of a thorough morphological and molecular analysis of the specimens of A. rapae from different regions of the world. In addition, it covers all significant references and thus provides an adequate discussion. I am sure that the authors presented extensive information on the subject, which supports their conclusions, and I therefore believe that the manuscript could be published in Insects. However, I suggest that this species should be called Aphidius rapae throughout the whole text, beginning from the title (except for the historical review), which eliminates the unnecessary ambiguity. I have also noticed a few possible misprints and similar errors (please see the attached file).

Comments on the Quality of English Language

As I noted above, the paper is generally well-written. Nevertheless, I would suggest a careful grammatic revision of the text, especially when it comes to using appropriate articles and prepositions. 

Reviewer 2 Report

Comments and Suggestions for Authors

The manuscript presents a comprehensive taxonomic revision of Diacretiella rapae, integrating morphological and molecular evidence to synonymize it under Aphidius as Aphidius rapae. The study is well-structured, methodologically rigorous, and addresses a historically contentious taxonomic issue.

Explicitly state why resolving D. rapae’s taxonomy is critical for biocontrol applications (e.g., implications for host-parasitoid databases or field efficacy). 

In discussion part, expand on why wing vein reduction is homoplastic in Aphidiinae and how this aligns with broader evolutionary trends in Hymenoptera. 

Reviewer 3 Report

Comments and Suggestions for Authors

The paper provided information about a large sample of a species that have been subject to debate. After their analysis using morphological measurements, evolutionary distances, and Bayesian analysis, authors point to a genus change for this species. Even though the results of the molecular data are suggestive of that conclusion, I believe that a more structured approach can be done.

The morphological data can be questioned as only data from the proposed genus to move the species to are provided. The subject species fits within the variation of the genus but not information on the genus Diaeretiella is provided and thus, there is no way to check whether these characters are diagnostic or not. The results section for this part is barely developed despite citing an incredible large number of specimens and measurements, the topic is closed with a single statement that does not offer any analytical evidence, no single statistical analysis of measurements. The diagnosis refers to a character that is questioned in several parts of the paper (wing vein reductions) and thus, does not allow a proper diagnosis of the species.

A redescription is given without information about the type specimen.

Specific comments

Introduction

Lines 40-43 I suggest connecting the two sentences with an and as these are on the same topic.

Lines 47-48. “Most considered Curtis to be the author of D. rapae in his book or in McIntosh's book, while only some people ascribed 48 authorship to McInthosh.” Please check the writing of this idea, the species was described either by one or another author,

There is no formal statement on the main goal of the paper

Materials and Methods

Lines 83. In this study is not necessary as you are describing the methods of your study.

Lines 86-87. Please explain the role of the other species. Describe what information was gathered

Lines 91-98. If possible, I suggest adding a schematic drawing with the measurements indicated

No Diaeretiella species are included!

Results

Lines 153. Number of antennomeres

Lines 190-192. Please consider rewriting “The most important character that currently discriminates genus Diaeretiella from 190 Aphidius is reduced wing venation with the absent median (M+m-cu) and r-m veins in D. 191 rapae,”, I suggest “The most important character that currently discriminates the genus Diaeretiella from Aphidius is the reduced wing venation with the median (M+m-cu) and r-m veins absent in D. rapae

Lines 202-203. Please provide more information about the more distant haplotypes such as geographic origin or hosts, if available.

Legend figure 7 lines 211-216. I believe this section should be part of the main text, not as part of the legend.

Lines 241-243. The entire morphological dataset is summarized in a statement that does not provide any analytical result or direct evidence.

Lines 246-248. The character provided as diagnostic is actually questioned by the same authors and no other characters are given to solve the situation.

Lines 257-265. A redescription is given without information about the type specimen.

Discussion

Lines 318-319. Authors claim that morphological characters prove that the species belong to Aphidius rather than to Diaeretiella but no comparative information is given in the latter to have a thorough comparison.

Lines 320-321. Please consider rewriting. A more standard statement can be “these results agree with DDD and DDD…” Hafez and Mackauer do not publish attitudes but statements or conclusions or hypothesis.

Lines 321-322. Authors stated that wing venation can not be used for id, however, the diagnosis describes characters related to wing venation absences.

Comments on the Quality of English Language

The english is good in general but in order to improve clarity, it can be improved through the help of a native speaker with knowledge in biological sciences

Round 2

Reviewer 3 Report

Comments and Suggestions for Authors

Authors have significantly improved their paper with only grammar or style adjustments required to have the paper ready for publication. These are my new comments:

A legend to the second figure is required and the legend must be cited. Parts and conventions must be described. Please check figure numbers and changes along the paper. You may move text of lines 103-107 to become the legend.

Lines 88-90. Please check English. “In addition, we examined specimens belonging to 43 Aphidius species listed in Table 1 as those species represent all variability of morphological characters in genus Aphidius.”. it may be as “In addition, we examined specimens belonging to 43 Aphidius species listed in Table 1 as those exhibit the entire morphological variability in genus Aphidius.”

Line 135. Please consider “To establish the phylogenetic position of Diaeretiella within subfamily Aphidiinae”

Legend figures 10 and 11. Please indicate what the squares are about

Line 282. Please check spaces between words “However, based only on wing venationspecimens”

Lines 405-406. Please check references to support the statement “There is a general trend within the Parasitica group of Hymenoptera toward a reduction in body size and wing venation. Wing” as there are studies challenging it. Rainford et al. (2016) challenges this idea for the Holometabola.

Comments on the Quality of English Language

please refer to the line comments
